# Gut Microbiota-Mediated Pharmacokinetic Drug–Drug Interactions between Mycophenolic Acid and Trimethoprim-Sulfamethoxazole in Humans

**DOI:** 10.3390/pharmaceutics15061734

**Published:** 2023-06-14

**Authors:** Nahathai Dukaew, Patcharawadee Thongkumkoon, Nutnicha Sirikaew, Sivamoke Dissook, Wannachai Sakuludomkan, Siripong Tongjai, Parameth Thiennimitr, Mingkwan Na Takuathung, Juthipong Benjanuwattra, Prachya Kongthaweelert, Nut Koonrungsesomboon

**Affiliations:** 1Department of Pharmacology, Faculty of Medicine, Chiang Mai University, 110 Intawaroros Road, Sriphoom, Muang, Chiang Mai 50200, Thailand; 2Clinical Research Center for Food and Herbal Product Trials and Development (CR-FAH), Faculty of Medicine, Chiang Mai University, Chiang Mai 50200, Thailand; 3Center of Multidisciplinary Technology for Advanced Medicine (CMUTEAM), Faculty of Medicine, Chiang Mai University, Chiang Mai 50200, Thailand; 4Department of Biochemistry, Faculty of Medicine, Chiang Mai University, Chiang Mai 50200, Thailand; 5Center for Research and Development of Natural Products for Health, Chiang Mai University, Chiang Mai 50200, Thailand; 6Department of Microbiology, Faculty of Medicine, Chiang Mai University, Chiang Mai 50200, Thailand; 7Research Center of Microbial Diversity and Sustainable Utilization, Chiang Mai University, Chiang Mai 50200, Thailand; 8Department of Internal Medicine, Texas Tech University Health Sciences Center, Lubbock, TX 79430, USA

**Keywords:** mycophenolate mofetil, mycophenolic acid, trimethoprim-sulfamethoxazole, pharmacokinetics, drug–drug interaction, gut microbiota

## Abstract

Mycophenolic acid (MPA) and trimethoprim-sulfamethoxazole (TMP-SMX) are commonly prescribed together in certain groups of patients, including solid organ transplant recipients. However, little is known about the pharmacokinetic drug–drug interactions (DDIs) between these two medications. Therefore, the present study aimed to determine the effects of TMP-SMX on MPA pharmacokinetics in humans and to find out the relationship between MPA pharmacokinetics and gut microbiota alteration. This study enrolled 16 healthy volunteers to take a single oral dose of 1000 mg mycophenolate mofetil (MMF), a prodrug of MPA, administered without and with concurrent use of TMP-SMX (320/1600 mg/day) for five days. The pharmacokinetic parameters of MPA and its glucuronide (MPAG) were measured using high-performance liquid chromatography. The composition of gut microbiota in stool samples was profiled using a 16S rRNA metagenomic sequencing technique during pre- and post-TMP-SMX treatment. Relative abundance, bacterial co-occurrence networks, and correlations between bacterial abundance and pharmacokinetic parameters were investigated. The results showed a significant decrease in systemic MPA exposure when TMP-SMX was coadministered with MMF. Analysis of the gut microbiome revealed altered relative abundance of two enriched genera, namely the genus *Bacteroides* and *Faecalibacterium*, following TMP-SMX treatment. The relative abundance of the genera *Bacteroides*, *[Eubacterium] coprostanoligenes* group, *[Eubacterium] eligens* group, and *Ruminococcus* appeared to be significantly correlated with systemic MPA exposure. Coadministration of TMP-SMX with MMF resulted in a reduction in systemic MPA exposure. The pharmacokinetic DDIs between these two drugs were attributed to the effect of TMP-SMX, a broad-spectrum antibiotic, on gut microbiota-mediated MPA metabolism.

## 1. Introduction

Mycophenolate mofetil (MMF), an ester-prodrug of mycophenolic acid (MPA), has been the antiproliferative immunosuppressant of choice for the prevention of graft rejection following solid organ transplantation [1,2]. It has also been increasingly used in some other conditions, such as hematopoietic stem cell transplantation and autoimmune diseases [3,4]. After oral administration, the prodrug MMF is rapidly hydrolyzed to its biologically active form, MPA, resulting in the first MPA peak plasma concentration. MPA is, then, mainly metabolized by glucuronidation in the liver to an inactive metabolite MPA-7-O-glucuronide (MPAG) and other minor metabolites. MPAG is partly excreted into bile [5,6,7], some of which is deconjugated back to MPA by gut microbial β-glucuronidase and is reabsorbed into blood circulation contributing to the second MPA peak in the MPA plasma concentration-time profile [6,7]. Due to its complex pharmacokinetics, MPA is prone to pharmacokinetic drug–drug interactions (DDIs) with several drugs, some of which are often used in solid organ transplant recipients and some other conditions in which MPA is indicated [7,8]. Significant pharmacokinetic DDIs between MPA and other coadministered drugs may alter MPA exposure, potentially resulting in undesirable outcomes, i.e., the lack of clinical efficacy or the development of adverse consequences [9,10].

The role of gut microbiota on MPA metabolism is well recognized [6,7], as the enterohepatic circulation is responsible for ~10–60% of overall systemic MPA exposure [11]. It is, therefore, perceivable that any medications significantly affecting the composition and/or metabolic activities of gut microbiota are likely to cause pharmacokinetic DDIs with MPA. Several studies suggest that quite a few drugs on the market may influence the human gut microbiota either by inhibiting gut bacterial proliferation or altering the composition of gut microbiota [12,13,14]. Theoretically, these drugs might have an impact on systemic MPA exposure by impairing the conversion of MPAG to MPA by gut bacterial β-glucuronidase. Antibiotics that exhibit significant antimicrobial activity against β-glucuronidase-producing anaerobic bacteria, such as norfloxacin and metronidazole, have been shown to predominantly affect the enterohepatic recycling of MPA, as evidenced by a significant reduction in the AUC_6–12_ or secondary peak of MPA [15].

Trimethoprim-sulfamethoxazole (TMP-SMX), a broad-spectrum antibiotic, is commonly prescribed for the prophylaxis of Pneumocystis pneumonia, one of the most common opportunistic infections in immunocompromised patients [16,17]. It is, thus, one of the most common drugs concurrently prescribed with MPA and other immunosuppressive agents during post-transplantation periods or in other MPA-indicated conditions [18]. TMP-SMX has been shown to affect the abundance, diversity, and composition of gut microbiota [19,20]. Additionally, it exhibits antibacterial activity against β-glucuronidase-positive bacteria [21]. Accordingly, the possible pharmacokinetic DDIs between MPA and TMP-SMX could be attributed to the antibiotic’s ability to eliminate certain β-glucuronidase-producing gut microbiota, some of which might play a role in the enterohepatic circulation of MPA. However, there has been little investigation into the possible pharmacokinetic DDIs between MPA and TMP-SMX; in other words, whether coadministration of TMP-SMX with MMF would result in a substantial change in systemic MPA exposure in humans is still uncertain [22]. Besides, there is still a gap of knowledge in terms of the specific types of gut microbes, of which their β-glucuronidase is responsible for the enterohepatic circulation of MPA.

Therefore, this study aimed at examining the effects of TMP-SMX on MPA pharmacokinetics in humans, as well as probable mechanisms of such a pharmacokinetic DDI by focusing on the relationship between MPA pharmacokinetics and gut microbiota alteration.

## 2. Materials and Methods

### 2.1. Study Design and Population

This prospective, open-label, fixed-sequenced study enrolled adult volunteers aged between 18 and 40 years with general good health based on medical history, physical examination, and basic laboratory tests (including, complete blood count, serum electrolytes, liver function test, and renal function test). Volunteers were excluded if they had a history of drug allergies or contraindication to MMF, TMP-SMX, or other sulfa agents (such as sulfonamide or probenecid), as well as those who had known or suspected pregnancy, chronic liver diseases, hematological disorders, or immunocompromised conditions. Individuals who had taken any antibiotics within 3 months prior to enrollment or regularly consumed alcohol-containing beverages, smoked cigarettes, or used an addictive substance were also disqualified. This study protocol was approved by the Research Ethics Committee of the Faculty of Medicine, Chiang Mai University (No. 102/2020 and 048/2021), and the Institutional Biosafety Committee of the Faculty of Medicine, Chiang Mai University (No. CMUIBC02012/2564). This study was prospectively registered at the Thai Clinical Trials Registry (TCTR20200518005), and it was conducted in accordance with the Declaration of Helsinki, the International Conference on Harmonization Guidelines for Good Clinical Practice (ICH-GCP), as well as local applicable regulations. Sixteen study participants provided written informed consent.

### 2.2. Drug Administration and Sample Collection

The study schema is depicted in Appendix A. In the first period, study participants received a single dose of 1000 mg MMF (Cellcept^®^; F. Hoffmann-La Roche Ltd., Basel, Switzerland) on Day 0. After a washout period of 9 days, study participants took 160/800 mg TMP-SMX (Bactrim^®^; Patar Lab (2517) Co., Ltd., Pathum Thani, Thailand), twice daily, on Days 10–14. In the second period, study participants received a single dose of 1000 mg MMF on Day 14. All the participants were required to abstain from alcohol-containing beverages, cigarette smoking, grapefruit juice, and any other medicinal products during the study period.

Blood samples were drawn at pre-dose and 0.33, 0.67, 1, 1.5, 2, 4, 6, 8, 10, 12, 24, 36, and 48 h post-MMF administration in both periods and kept in ethylenediaminetetraacetic acid disodium salts containing tubes. After that, the plasma was separated and stored at −20 °C until analysis. Stool samples were collected from the participants on Day 0, Day 9, and Day 14, using a standard stool collection kit. Three episodes of stool sampling were indicated as follows: Day 0 = pretreatment, Day 9 = post-MMF treatment but pre-TMP-SMX treatment, and Day 14 = post-TMP-SMX treatment. The stool collection on Day 9 was intended to be used as a confirmation ensuring that MMF treatment alone on Day 0 did not alter the gut microbiota. As a result, any differences in gut microbiota between Day 0 and Day 14 were reasonably assumed to be the effect of TMP-SMX treatment on Day 10–14. The stool samples were stored at −80 °C in RNAlater^TM^ reagent before DNA extraction.

### 2.3. Determination of MPA and MPAG

Plasma concentrations of MPA and MPAG were measured using high-performance liquid chromatography (HPLC) following a modified approach of the previously described method [23]. The carboxy butoxy ether mycophenolic acid (MPAC) was used as the internal standard in this study. The separation of MPA, MPAG, and the internal standard was carried out using Agilent Zorbax SB-C18 Analytical HPLC Column (4.6 × 250 mm) and an Agilent SB-C18 Guard Cartridge (4.6 × 12.5 mm). The mobile phase consisted of 20 mM KH_2_PO_4_ (pH 2.7, adjusted with 85% phosphoric acid) (A) and acetonitrile (B). The gradient conditions were as follows: 0–3 min (73% A), 3–4 min (73–50% A), 4–10 min (50–30% A), and 10–15 min (73% A). The flow rate was maintained at 1.2 mL/min with a column temperature of 45 °C. The UV detection wavelength was 254 nm. The standard curves for MPA and MPAG were linear over the range of 0.25–128 μg/mL. To assess the accuracy and precision, within-run and between-run measurements of MPA and MPAG concentrations were performed on low (2 μg/mL), medium (16 μg/mL), and high (64 μg/mL) quality control (QC) samples with 3 replicates for each, using calibration curves. Precision was expressed using the coefficient of variation (% CV). The concentrations of MPA and MPAG were found to be linear and accurate within the range of analysis with a CV % of less than 15%.

### 2.4. Pharmacokinetic Analysis

Pharmacokinetic parameters for MPA and MPAG were estimated by non-compartmental analysis using PKanalix 2020R1 software (Lixoft, Antony, France). Values of maximum concentration (C_max_) and time to reach maximum concentration (T_max_) were taken from the observed concentration–time curve data. Individual estimates of the apparent terminal elimination rate constant (λ_z_) were obtained from the logarithmic-linear regression of terminal portions of the plasma concentration–time curves. The area under the concentration–time curve (AUC) of pre-specified time points (including AUC_0–6_, AUC_6–12_, and AUC_0–12_) and AUC from the time of dosing (time 0) to the time of last measurable concentration (AUC_last_) were calculated according to the linear trapezoidal rule. AUC from time 0 to infinity (AUC_0–∞_) was then estimated from AUC_last_ + C_last_/λ_z_. Moreover, apparent volume of distribution (V_z_/F), terminal elimination half-life (t_1/2_), and apparent clearance (CL/F) were also then calculated.

All pharmacokinetic parameters of MPA and MPAG which were estimated from the first period (a single dose of MMF alone) were compared with those from the second period (a single dose of MMF during concurrent use of TMP-SMX). Statistical differences in pharmacokinetic parameters between the two periods were tested with paired t-test or Wilcoxon signed-rank test, as appropriate. *p* < 0.05 was considered to indicate statistical significance. The statistical analyses were performed with SPSS Statistics software version 23 (IBM Corp., Armonk, NY, USA).

### 2.5. Bacterial DNA Extraction, PCR Amplification, and Sequencing

Total DNA from the stool samples of healthy volunteers was extracted using QIAamp^®^ Fast DNA Stool Mini Kit (Qiagen^®^, Hilden, Germany) following the manufacturer’s instructions. The quantity and quality of DNA were measured using a spectrophotometer (NanoDrop^®^, Thermo Fisher Scientific, Wilmington, DE, USA). The DNA samples were sent to the Novogene Bio Technology Co., Ltd. (Nanjing Sequencing Center, Nanjing, China) for the V4 amplicon region sequencing analysis. The V4 hypervariable region in the 16S rRNA gene was amplified using the specific forward primer 515F (5′-GTGCCAGCMGCCGCGGTAA-3′) and reverse primer 806R (5′-GGACTACHVGGGTWTCTAAT-3′). After PCR amplification, paired-end sequencing (2 × 250 bp) was performed using the Illumina NovaSeq 6000 platform, according to the standard instructions of the 16S genomic sequencing library preparation protocol.

### 2.6. Gut Microbiome Analysis

The quality of raw paired-end reads was checked using FastQC software version 0.11.9 [24]. The raw sequencing data were processed and analyzed with the Quantitative Insights Into Microbial Ecology 2 (QIIME2; version 2021.8.0) pipeline [25]. An alpha-diversity was assessed by calculating the Shannon index. The statistical significance of alpha-diversity between the two groups was examined using the Wilcoxon signed-rank test. Bray-Curtis dissimilarity, weighted UniFrac, and unweighted UniFrac distances were calculated to evaluate the beta-diversity of gut microbiota. The statistical significance of beta-diversity between the two groups was examined using the PERMANOVA test. The sequences were taxonomy classified against SILVA 16S rRNA gene reference database release 138 [26,27]. All results of 16s rRNA gene sequencing after taxonomy classification were assigned to each category of bacteria as phylum to genus level.

The relative abundance (%) of each bacterial genus was analyzed and expressed as median (interquartile range; IQR). The Wilcoxon signed-rank test was used to detect taxa with significantly different abundances between paired samples. Statistical corrections for multiple comparisons were performed using the original false discovery rate (FDR) method of Benjamini–Hochberg with desired false discovery rate (Q). The differential abundance test between groups was done using linear discriminant analysis effect size (LEfSe) to determine the feature (taxonomic units) with effect relevance [28]. A co-occurrence network analysis of the main contributors and other genera in each treatment group was constructed based on Spearman’s rank correlation coefficient [29]. Moreover, the correlations between the microbial abundances and pharmacokinetic parameters of MPA and MPAG were also analyzed using Spearman’s rank correlation coefficient and visualized as a correlation heatmap. Genera with an abundance of less than 1 percent were excluded from the correlation analyses. Correlation between two parameters was interpreted as “weak” (*r* < 0.40), “moderate” (*r* = 0.40–0.69), “strong” (*r* = 0.70–0.89), or “very strong” (*r* ≥ 0.90) [30]. RStudio software version 1.3.1093 was used for statistical analyses, plotting, and visualization. All statistical tests were two-tailed and a threshold for statistical significance was set at *p* < 0.05.

## 3. Results

### 3.1. Demographics of Study Participants

Sixteen healthy Thai volunteers (aged: 25 to 32 years; 62.5% male) were enrolled in this study; their baseline characteristics are shown in Table 1. All study participants had normal physical and laboratory values at enrollment. The whole-exome-based single nucleotide polymorphisms (SNPs) phylogenetic analyses (Appendix A) revealed no significant differences in the genetic background information of the genes involved in the human gut microbiota (Appendix A) and human immune system (Appendix A).

### 3.2. Effect of TMP-SMX on the Pharmacokinetic Parameters of MPA and MPAG

The mean plasma concentration–time profiles of MPA and MPAG following the administration of MMF alone and in concurrent use with TMP-SMX are illustrated in Figure 1. Following MMF administration, MPA peak plasma concentration was observed at 0.67 h for both periods. The MPA plasma concentration–time profile showed a small decrease in overall systemic MPA exposure when TMP-SMX was coadministered with MMF, as compared to MMF given alone. The second peak of MPA was observed between 6 and 12 h in both periods, with a slight decrease during the period of concomitant MMF-TMP-SMX intake, when compared to MMF alone (Figure 1A). Figure 1B shows the MPAG plasma concentration–time profile, which was rather different from its parent compound MPA. MPAG reached the peak plasma concentration slower than MPA and declined more gradually than MPA. A small increase in MPAG plasma concentrations between 8 and 24 h was observed following coadministration of TMP-SMX with MMF when compared to MMF alone.

The pharmacokinetic parameters of MPA and MPAG in both periods are presented in Table 2. We observed a significant decrease in MPA AUC_0–∞_ by ~27% when MMF was in concurrent use with TMP-SMX (39.45 ± 17.81 vs. 54.00 ± 27.33 µg·h/mL, *p* = 0.041), while MPAG AUC_0–∞_ was significantly increased by ∼33% (719.23 ± 137.91 vs. 542.71 ± 225.14 µg·h/mL, *p* = 0.016). MPA AUC_6–12_ was also decreased by ∼30% (from 5.45 ± 3.22 to 3.79 ± 3.18 µg·h/mL) when MMF was co-administrated with TMP-SMX, but this finding did not reach statistical significance (*p* = 0.082). Other MPA pharmacokinetic parameters were not significantly altered by the coadministration of TMP-SMX with MMF. Instead, MPAG CL/F was significantly decreased (2.31 ± 1.29 vs. 1.44 ± 0.28 L/h, *p* = 0.015) and, in turn, MPAG t_1/2_ was significantly increased (11.49 ± 4.39 vs. 18.35 ± 8.32 h, *p* = 0.020), when TMP-SMX was concurrently used with MMF.

### 3.3. Gut Microbiota Composition before and after TMP-SMX Treatment

Following bacterial DNA sequencing, pre-processing, and amplicon sequence variant (ASV) identification, the number of sequences for each fecal sample from the 16 study participants was between 91,835 and 147,842 reads, with a total of 15,236 different ASVs assigned across all samples. The analysis did not indicate any significant differences in gut microbiota between stool samples on Day 0 and Day 9 in terms of relative abundance and diversity indices (Appendix A), confirming that gut microbiota was not altered by a single dose of MMF on Day 0. Further analysis was, therefore, focused on the changes in gut microbiota between stool samples on Day 0 (pretreatment) and Day 14 (post-TMP-SMX treatment) exclusively.

Overall, the patterns of genus relative abundance profiles of gut microbiota in the 16 study participants showed some differences between pretreatment and post-TMP-SMX treatment (Figure 2). The bacterial alpha-diversity analysis using the Shannon index did not show any significant differences between the two groups (Figure 3A). However, a significant distinction in beta-diversity was observed between pretreatment and post-TMP-SMX treatment (unweighted Unifrac (qualitative), *p* = 0.0045; weighted Unifrac (quantitative), *p* = 0.0375), indicating that the bacterial community structure was changed after TMP-SMX treatment (Figure 3B).

The top 20 most abundant bacterial genera in the pretreatment and post-TMP-SMX treatment groups are shown in Appendix A. The genera ranking lists were changed following TMP-SMX treatment; the most abundant bacterial genus in the pretreatment group was *Bacteroides* (accounting for 18.03%), while *Prevotella* was ranked as the most abundant bacteria in the post-TMP-SMX treatment group (accounting for 12.54%). The top three enriched genera (i.e., *Bacteroides*, *Prevotella*, and *Faecalibacterium*) were analyzed for their differential relative abundance between pretreatment and post-TMP-SMX treatment. When compared to pretreatment, the relative abundance of *Bacteroides* was significantly decreased (18.03 ± 11.24% vs. 11.50 ± 5.79%, *p* = 0.016) after TMP-SMX treatment, while the relative abundance of *Faecalibacterium* was significantly increased (4.83 ± 3.23% vs. 9.26 ± 6.35%, *p* = 0.026). The relative abundance of *Prevotella* was not significantly changed (Figure 3C). Among 25 species in the *Bacteroides* genus identified from the participants’ microbiome, the *Bacteroides plebeius* was the most abundant *Bacteroides* spp. and was significantly decreased after TMP-SMX treatment (6.23 ± 4.91% vs. 2.97 ± 2.14%, *p* = 0.014), when compared to pretreatment (Appendix A).

### 3.4. Co-Occurrence Network of the Highly Abundant Genera in Gut Microbiota

A co-occurring of the 20 most abundant bacterial genera in the pretreatment versus post-TMP-SMX treatment groups was carried out based on Spearman’s rank correlation (Figure 4). The patterns of co-occurrence genera networks were marked differences between pretreatment and post-TMP-SMX treatment. Compared to the pretreatment samples, more negative relationships of the genera in the post-TMP-SMX treatment samples were noticeably observed. The co-occurrence analysis results were consistent with the reduction in beta-diversity found in the samples following TMP-SMX treatment (Figure 3B).

We, then, focused on the potential influence of the two main contributors, whose abundance was significantly changed following TMP-SMX treatment, on the overall gut microbiota. For pretreatment samples, the abundance of the enriched genus, *Bacteroides*, significantly showed a strong positive correlation with *Blautia* (*r* = 0.71, *p* = 0.002) and moderately positive correlations with several genera including *Alistipes* (*r* = 0.63, *p* = 0.009), *Roseburia* (*r* = 0.54, *p* = 0.031) and uncultured genus in family *Lachnospiraceae* (*r* = 0.50, *p* = 0.049). Moreover, the abundance of *Faecalibacterium* significantly showed strongly positive correlations with genera *Subdoligranulum* (*r* = 0.89, *p* < 0.001) and *Sutterella* (*r* = 0.74, *p* = 0.001) and moderately positive correlations with *[Eubacterium]* coprostanoligenes group (*r* = 0.54, *p* = 0.031), respectively (Appendix A). In contrast, after TMP-SMX treatment, the decreased abundance of the genus *Bacteroides* significantly exhibited moderately positive correlations with genera *Clostridia* UCG-014 (*r* = 0.54, *p* = 0.029) and *Muribaculaceae* (*r* = 0.52, *p* = 0.041), but it showed strongly and moderately negative correlations with genera *Prevotella* (*r* = −0.73, *p* = 0.001) and UCG-002 (family *Oscillospiraceae*) (*r* = −0.62, *p* = 0.011), respectively. Moreover, the increased abundance of the genus *Faecalibacterium* significantly exhibited a strong positive correlation with *[Eubacterium] eligens* group (*r* = 0.74, *p* = 0.001) and moderately positive correlations with genera UCG-002 (family *Oscillospiraceae*) (*r* = 0.61, *p* = 0.013), *Parabacteroides* (*r* = 0.57, *p* = 0.023) and *Fusicatenibacter* (*r* = 0.56, *p* = 0.024). In contrast, it demonstrated strongly negative correlations with genera *Clostridia* UCG-014 (*r* = −0.85, *p* < 0.001) and *Muribaculaceae* (*r* = −0.78, *p* < 0.001) as well as moderately negative correlations with uncultured genus in family *Lachnospiraceae* (*r* = −0.60, *p* = 0.015) and genus *Agathobacter* (*r* = −0.50, *p* = 0.047) (Appendix A).

### 3.5. Correlation between Gut Microbiota and the Pharmacokinetic Parameters of MPA and MPAG

The correlations between gut microbiota and MPA or MPAG pharmacokinetics were determined based on the bacterial abundance ratio of post-TMP-SMX treatment to pretreatment and the pharmacokinetic parameter ratio of the second period (MMF + TMP-SMX) to the first period (MMF alone) (Figure 5). The detailed results of Figure 5, Spearman’s coefficient (*r*), and *p*-values are summarized in Appendix A.

For MPA, the AUC_6–12_ ratio exhibited moderately positive correlations with the genera *Bacteroides*, *[Eubacterium] eligenes* group, and *[Eubacterium] coprostanoligenes* group abundance ratios (*r* = 0.67, *p* = 0.004; *r* = 0.61, *p* = 0.012; *r* = 0.52, *p* = 0.041, respectively). In addition, the AUC_0–∞_ ratio exhibited moderately positive correlations with the genera *Ruminococcus* and *Bacteroides* abundance ratios (*r* = 0.53, *p* = 0.035; *r* = 0.52, *p* = 0.041, respectively). In contrast, the abundance ratios of these two bacteria were significantly negatively correlated with the CL/F ratio (*r* = −0.53, *p* = 0.035; *r* = −0.52, *p* = 0.041, respectively).

For MPAG, the genera *Bacteroides*, *[Eubacterium] coprostanoligenes* group, and *[Eubacterium] eligenes* group abundance ratios were significantly negatively correlated with the AUC_0–∞_ ratio (*r* = −0.55, *p* = 0.026; *r* = −0.62, *p* = 0.011; *r* = −0.59, *p* = 0.017, respectively), but positively correlated with the CL/F ratio (*r* = 0.55, *p* = 0.026; *r* = 0.62, *p* = 0.011; *r* = 0.59, *p* = 0.017, respectively). The genera *Muribaculaceae* and *Ruminococcus* abundance ratios markedly appeared to have significantly negative correlations with the ratio of several MPAG pharmacokinetic parameters, including AUC_0–12_ (*r* = −0.70, *p* = 0.003; *r* = −0.68, *p* = 0.004, respectively), AUC_6–12_ (*r* = −0.66, *p* = 0.005; *r* = −0.67, *p* = 0.005, respectively), and AUC_0–6_ (*r* = −0.52, *p* = 0.037; *r* = −0.56, *p* = 0.024, respectively). The genera *Muribaculaceae* and *Ruminococcus* abundance ratios were also significantly positively correlated with the V_z_/F ratio (*r* = 0.63, *p* = 0.009; *r* = 0.61, *p* = 0.012, respectively).

## 4. Discussion

The present study observed a significant reduction in MPA AUC_0–∞_ by around 27% when TMP-SMX was concurrently administered with MMF in adult volunteers who are in good health. After concurrent dosing, an approximate 30% reduction in MPA AUC_6–12_ was also observed, whereas other MPA pharmacokinetic parameters were somewhat identical between the two periods. Based on our observation, it may be reasonable to postulate that oral administration of TMP-SMX, a broad-spectrum antibiotic, might predominantly affect the enterohepatic circulation of MPA, thereby reducing systemic MPA exposure, while the primary absorption of MPA following MMF administration was not significantly altered. The possible pharmacokinetic DDI mechanism of TMP-SMX and MPA may be due, at least in part, to the TMP-SMX’s action on the alteration of β-glucuronidase-producing gut microbes, thereby suppressing the conversion of MPAG back to MPA in the intestine. Previous studies also noted a significant change in MPA pharmacokinetics following oral administration of some other antibiotics, such as ciprofloxacin [31,32,33], norfloxacin [15], metronidazole [15,32], amoxicillin/clavulanic acid [32,33,34], tobramycin plus cefuroxime [35], and vancomycin [36]. Most studies found that these antibiotics significantly reduce systemic MPA exposure while having little impact on MPA primary absorption [15,31,32,33,34,35].

In this study, we observed a significant alteration in the gut microbial community structure on the genus level after TMP-SMX treatment in the volunteers. The relative abundance of some enriched genera, such as *Bacteroides*, was substantially altered after TMP-SMX treatment, and such an alteration was shown to be significantly correlated with a reduction in systemic MPA exposure. The literature supports our results that the genus *Bacteroides* is one of the likely key contributors to gut microbiota-mediated MPA metabolism. First, the genus *Bacteroides* has recently been identified as the main contributor to the gut β-glucuronidase pool, encoding ~55% of all β-glucuronidase enzymes [37,38]. Among 200 genera previously identified from the human microbiome [39], only the genus *Bacteroides* exhibited a strong correlation with the β-glucuronidase level [40]. Second, the genus *Bacteroides* encodes the main mL1- and some L1-β-glucuronidases, two of which are the major types of β-glucuronidases in breaking down drug-glucuronide conjugates in the gastrointestinal tract [31,37,41,42,43]. Third, it has been recently observed that hematopoietic cell transplant patients with high MPA enterohepatic circulation have a significantly greater abundance of *Bacteroides* species in stool than those with low MPA enterohepatic circulation [44]. Besides the genus *Bacteroides*, the genera *[Eubacterium] coprostanoligenes* group, *[Eubacterium] eligens* group, and *Ruminococcus* of the phylum Firmicutes also appeared to be significantly correlated with systemic MPA exposure. Some bacterial species in these genera, such as *E. eligens* and *R. gnavus*, are characterized as the L1 β-glucuronidases-encoding bacteria [45,46]. Considering our findings and the literature, some bacterial genera/species may play an important role in gut microbiota-mediated MPA metabolism.

Coadministration of TMP-SMX with MMF altered not only the pharmacokinetics of MPA but also MPAG, as shown by significant increases in MPAG AUC_0–∞_ and MPAG t_1/2_, and a decrease in MPAG CL/F. This observation alerted us to the possibility of other pathways involved in pharmacokinetic DDIs between TMP-SMX and MPA. For instance, the excretion of MPAG involves several transport mechanisms including P-glycoprotein (P-gp), breast cancer resistance protein (BCRP), multi-drug resistant protein 2 (MRP2), and organic anion-transporting polypeptide (OATP) [9,47]. These transporters potentially have a significant overlap of inhibitors with the bile salt export pump (BSEP/ABCB11) [48], which might be inhibited by sulfamethoxazole [49]. Therefore, it is not outside the realm of possibility that the inhibition of some transporters responsible for the excretion of MPAG may play a role in TMP-SMX-MPA/MPAG interactions. However, it is too early to draw any conclusions based on the available evidence. Further investigation is required to understand more specific mechanisms attributable to TMP-SMX-MPA/MPAG interactions other than gut microbiota-mediated MPA metabolism.

Lastly, the study’s findings should be interpreted in light of its limitations. First and foremost, the β-glucuronidase activity in stool samples, as well as the levels of MPA and MPAG in the intestinal lumen, were not measured in this study. Hence, whether the alteration of β-glucuronidase-producing bacteria after TMP-SMX treatment as observed in this study would significantly affect the bacterial β-glucuronidase activity and/or MPA-MPAG metabolism in the intestine was not confirmed. Second, the present investigation was conducted in healthy volunteers, whose gut microbiota composition may differ from some other populations with certain diseases/conditions [50,51]. Further research may be necessary to determine the clinical significance and implication of the findings in those populations.

## 5. Conclusions

In conclusion, the coadministration of TMP-SMX with MMF could result in a decrease in systemic MPA exposure in humans. The pharmacokinetic DDI between these two medications might be attributable, at least in part, to the broad-spectrum antibiotic’s action altering gut microbiota-mediated MPA metabolism. TMP-SMX altered the composition of gut microbiota, notably by lowering the relative abundance of the major β-glucuronidase-producing bacteria, such as the genus *Bacteroides*. Our findings suggest that clinicians should be aware of potential pharmacokinetic DDIs when administering TMP-SMX during MMF therapy. Further studies are required to determine any clinical relevance of such pharmacokinetic DDIs in some specific groups of the population, such as solid organ transplant recipients, for whom these two medications are frequently prescribed together.

## Figures and Tables

**Figure 1 pharmaceutics-15-01734-f001:**
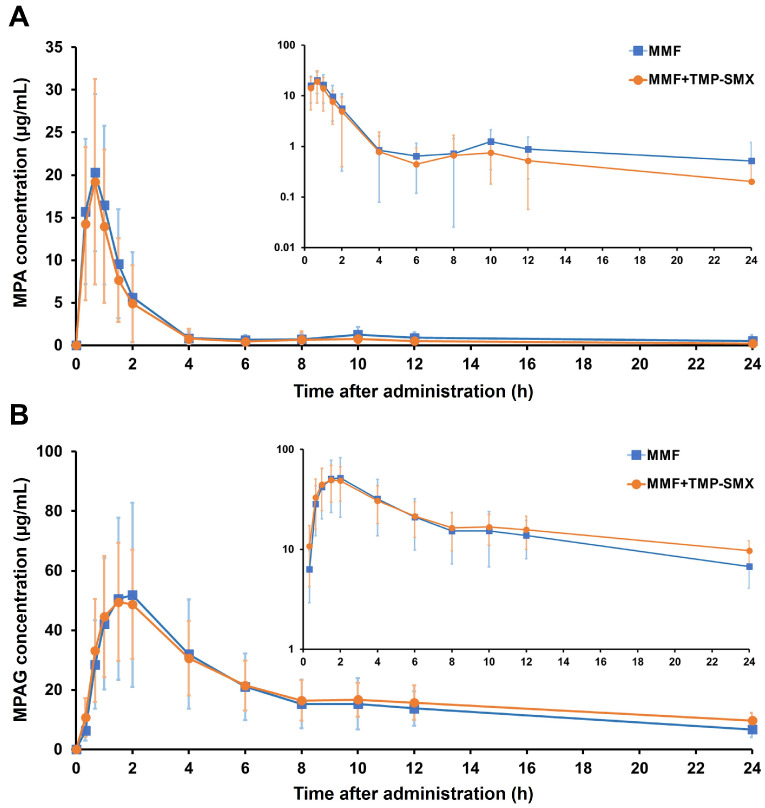
Concentration–time curves of MPA (**A**) and MPAG (**B**) following MMF administration with or without TMP-SMX treatment. The semilog scale of each concentration–time curve is inserted. Values are means ± SD.

**Figure 2 pharmaceutics-15-01734-f002:**
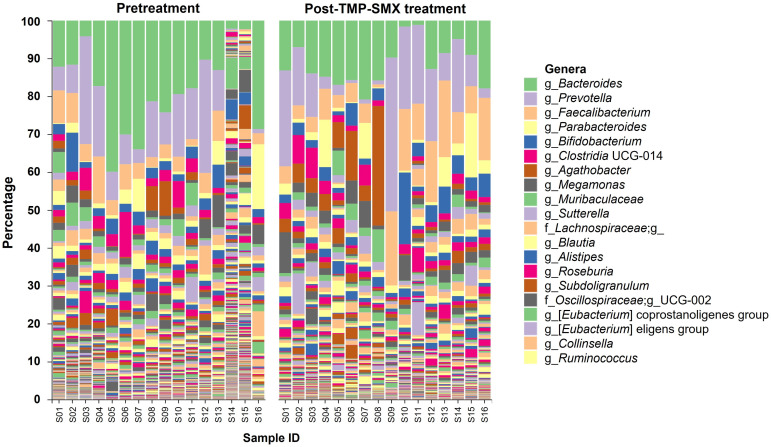
Genus relative abundance profiles of gut microbiota between pretreatment and post-TMP-SMX treatment. The labels on the right show the top 20 most abundant genera across the samples.

**Figure 3 pharmaceutics-15-01734-f003:**
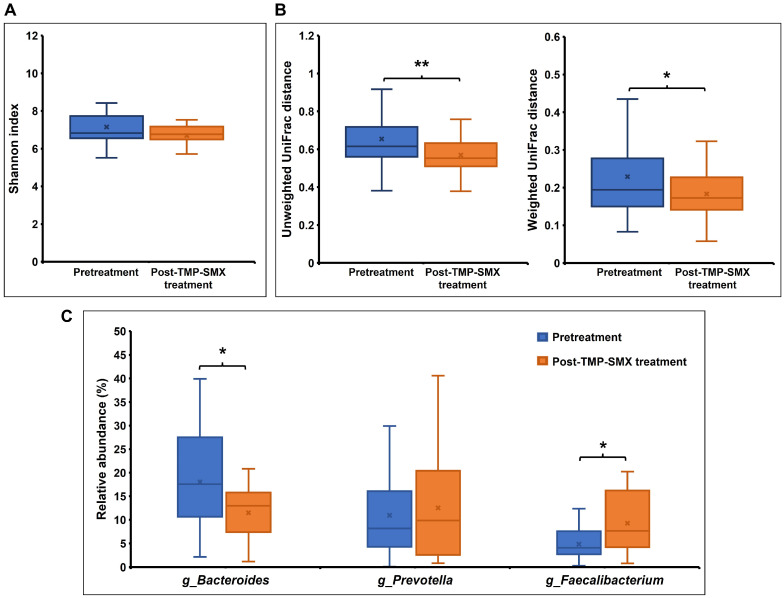
Alteration of the gut microbiota composition after TMP-SMX treatment. (**A**) Boxplot of gut microbial alpha-diversity (Shannon index); (**B**) Boxplot of gut microbial beta-diversity (Unweighted and weighted UniFrac distance). The box represents the interquartile range (IQR) between the first and third quartiles, and the line and cross (×) inside the box represent the median and mean, respectively. (**C**) Relative abundance of the three main abundant genera in the 16 study participants between pretreatment and post-TMP-SMX treatment. *p*-value determined the significant difference after statistical corrections using the original false discovery rate (FDR) method of Benjamini–Hochberg for multiple comparisons (Q), * *p* < 0.05; ** *p* < 0.01.

**Figure 4 pharmaceutics-15-01734-f004:**
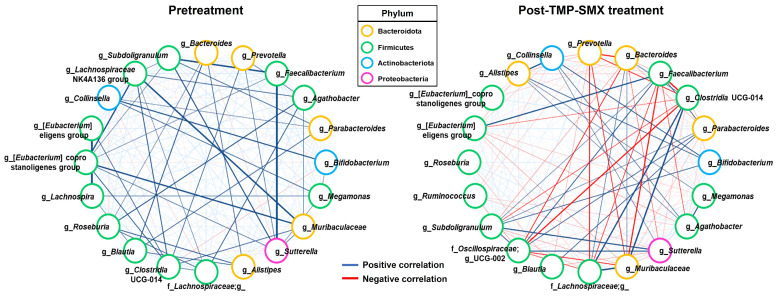
Co-occurrence network of the highly abundant genera in gut microbiota at pretreatment and post-TMP-SMX treatment. Blue lines represent positive correlations, while red lines represent negative correlations; the thicker the line is, the stronger the Spearman’s rank correlation is. Only the top 20 most abundant genera of each group were included in this analysis. The border color of each node indicates the phylum.

**Figure 5 pharmaceutics-15-01734-f005:**
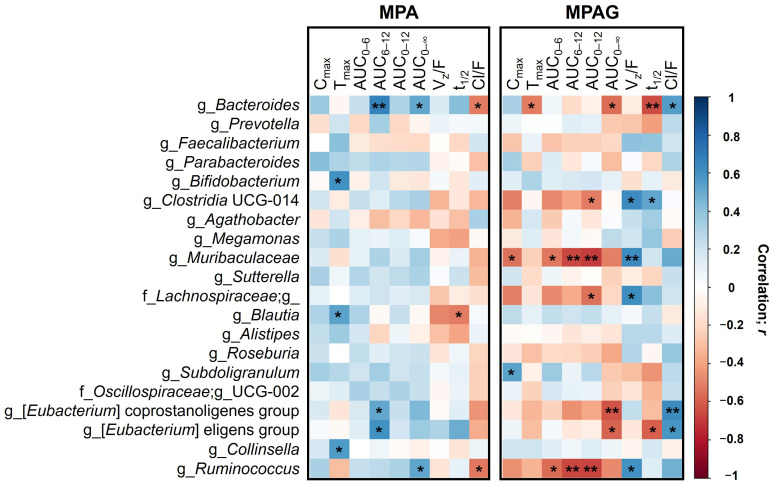
Correlation heatmap between the gut microbial abundance and the pharmacokinetics of MPA and MPAG. A heatmap shows the correlations between the bacterial abundance ratio of post-TMP-SMX treatment to pretreatment (rows) and the MPA or MPAG pharmacokinetic parameters ratio of the second period (MMF + TMP-SMX) to the first period (MMF alone) (columns). Positive correlations are illustrated using a gradient of blue colors, whereas negative correlations are illustrated using a gradient of red colors; the darker the color is, the stronger the Spearman’s rank correlation is. Correlations with *p*-values that are statistically significant are noted by an asterisk (*); * *p* < 0.05 and ** *p* < 0.01. Only the top 20 most abundant genera across the samples were included in this analysis.

**Table 1 pharmaceutics-15-01734-t001:** Baseline characteristics of the study participants.

Characteristics	Value (*n* = 16)
Sex, *n* (%)	
Male	10 (62.5%)
Female	6 (37.5%)
Age (year)	28.88 ± 2.66
Weight (kg)	59.19 ± 9.57
BMI (kg/m^2^)	21.52 ± 2.62
Heart rate (beats/min)	76.94 ± 11.35
Blood pressure; Systolic/Diastolic (mmHg)	118.63 ± 11.98/70.31 ± 8.61
Laboratory parameters	
Complete blood count	
Hemoglobin (g/dL)	13.10 ± 1.15
Hematocrit (%)	40.99 ± 3.88
White blood cell count (cells/μL)	6346.25 ± 1028.87
Platelets (cells/μL)	282,937.50 ± 55,789.45
Serum electrolytes	
Na^+^ (mmol/L)	138.06 ± 1.69
K^+^ (mmol/L)	3.94 ± 0.22
Cl^−^ (mmol/L)	100.44 ± 1.90
HCO_3_^−^ (mmol/L)	23.50 ± 1.46
Liver function test	
Albumin (g/dL)	4.55 ± 0.16
Total bilirubin (mg/dL)	1.03 ± 1.35
Direct bilirubin (mg/dL)	0.40 ± 0.28
ALT (U/L)	17.25 ± 9.06
AST (U/L)	20.56 ± 6.41
ALP (U/L)	59.50 ± 17.68
Renal function test	
BUN (mg/dL)	12.31 ± 2.85
Creatinine (mg/dL)	0.85 ± 0.19
eGFR (mL/min/1.73 m^2^)	109.78 ± 13.14

Categorical variables are expressed as numbers (percentages). Continuous variables are expressed as mean ± standard deviation. Abbreviations: ALP, alkaline phosphatase; ALT, alanine aminotransferase; AST, aspartate aminotransferase; BMI, body mass index; BUN, blood urea nitrogen; eGFR, estimated glomerular filtration rate.

**Table 2 pharmaceutics-15-01734-t002:** Pharmacokinetic parameters of MPA and MPAG following MMF administration with and without TMP-SMX treatment.

Parameters	Mean ± Standard Deviation		*p* Value ^b^
	MMF	MMF + TMP-SMX	
**MPA**			
C_max_ (µg/mL)	25.89 ± 8.65	23.41 ± 9.99	0.446
T_max_ (h) ^a^	0.67 (0.33–1.5)	0.67 (0.33–2)	0.942
AUC_0–6_ (µg·h/mL)	33.03 ± 13.17	28.98 ± 12.38	0.227
AUC_6–12_ (µg·h/mL)	5.45 ± 3.22	3.79 ± 3.18	0.082
AUC_0–12_ (µg·h/mL)	38.48 ± 14.71	32.76 ± 13.31	0.119
AUC_0-∞_ (µg·h/mL)	54.00 ± 27.33	39.45 ± 17.81	0.041 *
V_z_/F (L)	166.29 ± 69.83	161.58 ± 61.40	0.853
t_1/2_ (h)	6.25 ± 3.91	4.47 ± 2.98	0.122
CL/F (L/h)	22.51 ± 9.69	36.63 ± 34.27	0.104
**MPAG**			
C_max_ (µg/mL)	54.87 ± 30.36	52.67 ± 18.61	0.704
T_max_ (h) ^a^	1.5 (1–2)	1.5 (1–2)	0.963
AUC_0–6_ (µg·h/mL)	204.36 ± 110.23	201.57 ± 74.45	0.890
AUC_6–12_ (µg·h/mL)	96.04 ± 49.30	103.50 ± 37.46	0.582
AUC_0–12_ (µg·h/mL)	300.39 ± 157.32	305.07 ± 101.75	0.887
AUC_0-∞_ (µg·h/mL)	542.71 ± 225.14	719.23 ± 137.91	0.016 *
V_z_/F (L)	36.32 ± 17.51	37.48 ± 16.77	0.797
t_1/2_ (h)	11.49 ± 4.39	18.35 ± 8.32	0.020 *
CL/F (L/h)	2.31 ± 1.29	1.44 ± 0.28	0.015 *

^a^ T_max_ is presented as the median (min-max). ^b^ *p*-value was obtained by the paired *t*-test or the Wilcoxon signed-rank test: * *p* < 0.05. Abbreviations: AUC, area under the concentration–time curve (with pre-specified time points); AUC_0-∞_, AUC from time 0 to infinity; CL/F, apparent clearance; C_max_, maximum concentration; MMF, mycophenolate mofetil; MPA, mycophenolic acid; MPAG, mycophenolic acid-7-O-glucuronide; t_1/2_, terminal elimination half-life; T_max_, time to reach maximum concentration; TMP-SMX, trimethoprim-sulfamethoxazole; V_z_/F, apparent volume of distribution.

## Data Availability

The data used during the current study, including 16S rRNA sequencing data, will be available in the NCBI Sequence Read Archive (SRA) database under the Bioproject ID PRJNA932165 (https://www.ncbi.nlm.nih.gov/sra/PRJNA932165) (accessed on 8 February 2023). This paper does not report any original code. Any additional information required to reanalyse the data reported in this paper is available from the corresponding author upon reasonable request.

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
