# Peer review of "Gut Microbiota-Mediated Pharmacokinetic Drug–Drug Interactions between Mycophenolic Acid and Trimethoprim-Sulfamethoxazole in Humans"

_pharmaceutics, 2023, doi:10.3390/pharmaceutics15061734_

Round 1
Reviewer 1 Report
Immunosuppressive drugs could alter the composition of the gut microbiome, which could influence the metabolism of immunosuppressive drugs and the immune system of transplant patients. The gut microbiome offers a new opportunities for precision medicine in transplantation. The relationship exists also between mycophenolic acid and the gut microbiome and suggest that the gut microbiome has a strong contribution to the pathophysiology of mycophenolic acid-induced enteropathy. A full description of the bidirectional interaction between the gut microbiome and immunosuppressive drugs, in general, should document both the ability of the drug to induce dysbiosis and the changes in the metabolic profile of the drug induced by the gut microbiote.
A perfect example to illustrate this interaction between the immunosuppressive drugs and the gut microbiome is the case of mycophenolic acid (MPA), active metabolite of mycophenolate mofetil. MPA, originally isolated as a fermentation product of Penicillium species, is a broad-spectrum drug with antibacterial, antifungal and antiviral properties, in addition to its immunosuppressive properties. This pharmacological profile may explain its ability to modify the microbial composition and metabolism. Another key element is the contribution of the gut microbiome to MPA metabolism. Regarding the metabolic profile MPA inactivation occurs primarily in the liver through glucuronidation, where MPA is conjugated to glucuronic acid. This leads to the production of the major metabolite mycophenolic acid glucuronide. While the majority of it is excreted in urine, the remainder is is excreted in the bile. Once excreted, MPAG interacts with commensal gut bacteria in the lower gastrointestinal tract where bacterial β-glucuronidase hydrolyzes MPAG back to its active form, MPA. MPA in turn interacts with the intestinal epithelium and undergoes enterohepatic recirculation which contributes to 30 to 40 % of the plasma MPA concentration. Local accumulation of MPA close to digestive epithelial cells is thought to initiate severe gastrointestinal adverse symptoms such as nausea, vomiting, abdominal pain and diarrhea associated with MPA therapy, a typical drug-induced enteropathy characterized by architectural disorganization of the gastrointestinal epithelium with edema and hemorrhagic ulceration.
In this context, the submitted paper is aimed on gut microbiota-mediated pharmacokinetic drug-drug interactions mycophenolic acid and trimethoprim-sulfamethoxazole in humans. I read throughout the manuscript. The manuscript is well-written, the figures are legible and appropriate to the manuscript. However, my major concern is focused on the problem, which describes by the authors themselves as limitation of the study, namely that the present investigation was done in healthy volunteers, whose gut microbiota composition may differ from population of transplant patients. Therefore, it seems to me that authors post in manuscript a far-going conclusions, In my opinion this study presents just only a half of the study including only a control group.
Based on the above observations, I think that the manuscript in its current form is not sufficient for publication and I recommend to re-submission it after supplementing a research group of patients. Moreover, there are more comments of the paper concerning HPLC method for determination of MPA and MPAG. Authors noted, that analytical method used is modified approach of the previously described method for PK studies in rat plasma and bile published in 2011. Thera are a lots of published , validated method for MPA and MPAG analysis, more than 20 in the literatures. Why the authors did not present the validation parameters of the method? It is necessary to make sure that results are correct. What the substance was used as internal standard? Why the concentration range of calibration curve was very wide up to 128 µg/mL? While, in case of MPA is sufficient up to 50 µg/mL, otherwise in case of MPAG f.e. in transplant patients with renal impairment up to 500 µg/mL.
Author Response
We are thankful for the careful and insightful review and editing of our manuscript. We sincerely appreciate all the valuable comments and suggestions, which have helped us improve the quality of our manuscript. Our responses to the comments are described below in a point-by-point manner. We have incorporated the suggested changes made by the reviewers into the manuscript using Track Changes. In any case, we remain open to further comments and suggestions. Thank you for your kind consideration.
Reviewer 1’s comment #1:
Immunosuppressive drugs could alter the composition of the gut microbiome, which could influence the metabolism of immunosuppressive drugs and the immune system of transplant patients. The gut microbiome offers a new opportunities for precision medicine in transplantation. The relationship exists also between mycophenolic acid and the gut microbiome and suggest that the gut microbiome has a strong contribution to the pathophysiology of mycophenolic acid-induced enteropathy. A full description of the bidirectional interaction between the gut microbiome and immunosuppressive drugs, in general, should document both the ability of the drug to induce dysbiosis and the changes in the metabolic profile of the drug induced by the gut microbiote.
A perfect example to illustrate this interaction between the immunosuppressive drugs and the gut microbiome is the case of mycophenolic acid (MPA), active metabolite of mycophenolate mofetil. MPA, originally isolated as a fermentation product of Penicillium species, is a broad-spectrum drug with antibacterial, antifungal and antiviral properties, in addition to its immunosuppressive properties. This pharmacological profile may explain its ability to modify the microbial composition and metabolism. Another key element is the contribution of the gut microbiome to MPA metabolism. Regarding the metabolic profile MPA inactivation occurs primarily in the liver through glucuronidation, where MPA is conjugated to glucuronic acid. This leads to the production of the major metabolite mycophenolic acid glucuronide. While the majority of it is excreted in urine, the remainder is is excreted in the bile. Once excreted, MPAG interacts with commensal gut bacteria in the lower gastrointestinal tract where bacterial β-glucuronidase hydrolyzes MPAG back to its active form, MPA. MPA in turn interacts with the intestinal epithelium and undergoes enterohepatic recirculation which contributes to 30 to 40 % of the plasma MPA concentration. Local accumulation of MPA close to digestive epithelial cells is thought to initiate severe gastrointestinal adverse symptoms such as nausea, vomiting, abdominal pain and diarrhea associated with MPA therapy, a typical drug-induced enteropathy characterized by architectural disorganization of the gastrointestinal epithelium with edema and hemorrhagic ulceration.
Response:
Thank you very much for your thorough review and for bringing up the bidirectional interaction between the gut microbiome and immunosuppressive drugs. As the reviewer pointed out, previous studies have reported that more than 30% of patients on long-term maintenance with mycophenolate mofetil (MMF) as part of their immunosuppressive regimen experience gastrointestinal adverse effects, commonly known as mycophenolate-induced enteropathy [1,2]. It has been found that MMF-related gastrointestinal toxicity is a consequence of alterations in the composition of the gut microbiota [3,4] specifically through the selection of β-glucuronidase-expressing bacteria. The upregulation of β-glucuronidase expression and activity results in elevated concentrations of MPA in the colon, which in turn is associated with colonic inflammation [4]. Thus, it is crucial to consider the drug’s potential to induce changes in gut microbiota composition in individuals on long-term MMF maintenance.
However, the primary objective of this study was to investigate the pharmacokinetic drug-drug interaction (DDI) between MPA and trimethoprim-sulfamethoxazole (TMP-SMX), an antibiotic, in humans, as well as to explore the probable mechanisms underlying this pharmacokinetic DDI in relation to the alteration of gut microbiota composition induced by TMP-SMX. Our study design involved treating the participants with a single dose of MMF before and after TMP-SMX administration to demonstrate the pharmacokinetic DDI between these drugs. We were also concerned that a single dose of MMF treatment alone might influence the composition of the gut microbiota. Therefore, we designed the collection of stool samples from the participants into three distinct episodes: Day 0 (pretreatment), Day 9 (post-MMF treatment but pre-TMP-SMX treatment), and Day 14 (post-TMP-SMX treatment). The stool samples collected on Day 9 were intended to serve as a confirmation, ensuring that a single dose of MMF treatment on Day 0 did not cause any changes in the gut microbiota.
Our results, as shown in Supplementary Figure S4, did not reveal any significant differences in the gut microbiota between the stool samples obtained on Day 0 and Day 9, both in terms of relative abundance and diversity indices. This finding confirmed that the gut microbiota remained unaffected by a single dose of MMF on Day 0. As a result, any observed disparities in the gut microbiota between Day 0 and Day 14 were reasonably attributed to the effects of TMP-SMX treatment administered on Days 10–14, subsequently impacting MPA pharmacokinetics.
Reviewer 1’s comment #2:
In this context, the submitted paper is aimed on gut microbiota-mediated pharmacokinetic drug-drug interactions mycophenolic acid and trimethoprim-sulfamethoxazole in humans. I read throughout the manuscript. The manuscript is well-written, the figures are legible and appropriate to the manuscript. However, my major concern is focused on the problem, which describes by the authors themselves as limitation of the study, namely that the present investigation was done in healthy volunteers, whose gut microbiota composition may differ from population of transplant patients. Therefore, it seems to me that authors post in manuscript a far-going conclusions, In my opinion this study presents just only a half of the study including only a control group. Based on the above observations, I think that the manuscript in its current form is not sufficient for publication and I recommend to re-submission it after supplementing a research group of patients.
Response:
Thank you for your feedback. We fully acknowledge the reviewer’s point regarding the evaluation of the pharmacokinetic DDI between MPA and TMP-SMX in healthy volunteers rather than in the target patient population. It is true that there might be variations in gut microbiota composition between healthy volunteers and actual patient populations, potentially leading to differences in both pharmacokinetic and clinical results. Thus, we have included this limitation in our manuscript to ensure that readers are aware of this potential constraint.
However, in patient populations where MPA is commonly used, such as solid organ transplantation, individuals are often treated with multiple medications concurrently, including various immunosuppressants, proton pump inhibitors, and antibiotics. This makes it challenging to control intervention and confounding factors in order to prove the pharmacokinetic DDI between the MPA and TMP-SMX. Moreover, it would be unethical to discontinue either of the two drugs in order to study the pharmacokinetic DDI in these patient populations.
Furthermore, relevant international guidelines recommend that clinical DDI studies can be conducted on healthy volunteers, with the assumption that findings in healthy subjects can be extrapolated to the intended patient population [5,6]. Therefore, we initially designed the pharmacokinetic DDI study to be conducted on healthy volunteers. However, we acknowledge that further research is still necessary to confirm whether the observed pharmacokinetic DDI persists in the intended patient population. It is also important to determine the clinical significance and implications of these findings. These aspects should be investigated in future studies to provide a more comprehensive understanding of the pharmacokinetic and clinical implications of the MPA and TMP-SMX interaction.
Reviewer 1’s comment #3:
Moreover, there are more comments of the paper concerning HPLC method for determination of MPA and MPAG. Authors noted, that analytical method used is modified approach of the previously described method for PK studies in rat plasma and bile published in 2011. Thera are a lots of published, validated method for MPA and MPAG analysis, more than 20 in the literatures. Why the authors did not present the validation parameters of the method? It is necessary to make sure that results are correct. What the substance was used as internal standard? Why the concentration range of calibration curve was very wide up to 128 µg/mL? While, in case of MPA is sufficient up to 50 µg/mL, otherwise in case of MPAG f.e. in transplant patients with renal impairment up to 500 µg/mL.
Response:
Thank you for your comments and suggestions. We appreciate the reviewer’s recommendation to provide clearer details about the HPLC method in our manuscript. In this study, we analyzed the parameters to ensure that the procedure gives consistent and reliable results. For the HPLC analysis, we used carboxy butoxy ether mycophenolic acid (MPAC) as the internal standard compound. Standards curves for MPA and MPAG were constructed within a range of 0.25 to 128 µg/ml. The maximum concentration of MPA was found to be approximately 20 µg/ml, whereas the highest concentration of MPAG reached around 50 µg/ml. Ideally, the selection of a concentration that falls within the mid to upper range of the standard curve ensures it is well within the linear range. This approach provides the best accuracy and precision for quantifying the analyte concentration in the samples. In investigations employing the HPLC technique to examine mycophenolic acid content, a wide range for the standard curve is commonly employed, even when the quantity observed in the sample is comparatively lower [7,8]. The quality control (QC) assessment of the samples is used to determine the accuracy of this standard curve range. Precision is expressed using the coefficient of variation (CV%). Based on the experimental data shown in the table below, the results indicate reliable intra-day and inter-day dependability, ranging between 99% and 108%, and 97% and 106%, respectively. The corresponding % CVs were all below 15%. In the revised manuscript, we have added the precision and accuracy assessment method, as well as the information about the internal standard compound used, in the Materials and Methods section (2.3. Determination of MPA and MPAG).
References
[1] Behrend, M. Adverse gastrointestinal effects of mycophenolate mofetil: aetiology, incidence and management. Drug Saf 2001, 24, 645-663, doi:10.2165/00002018-200124090-00002.
[2] Al-Absi, A.I.; Cooke, C.R.; Wall, B.M.; Sylvestre, P.; Ismail, M.K.; Mya, M. Patterns of injury in mycophenolate mofetil-related colitis. Transplant Proc 2010, 42, 3591-3593, doi:10.1016/j.transproceed.2010.08.066.
[3] Jardou, M.; Provost, Q.; Brossier, C.; Pinault, É.; Sauvage, F.L.; Lawson, R. Alteration of the gut microbiome in mycophenolate-induced enteropathy: impacts on the profile of short-chain fatty acids in a mouse model. BMC Pharmacol Toxicol 2021, 22, 66, doi:10.1186/s40360-021-00536-4.
[4] Taylor, M.R.; Flannigan, K.L.; Rahim, H.; Mohamud, A.; Lewis, I.A.; Hirota, S.A.; Greenway, S.C. Vancomycin relieves mycophenolate mofetil-induced gastrointestinal toxicity by eliminating gut bacterial β-glucuronidase activity. Sci Adv 2019, 5, eaax2358, doi:10.1126/sciadv.aax2358.
[5] U.S. Department of Health and Human Services; Food and Drug Administration. Drug Interactions | Relevant Regulatory Guidance and Policy Documents FDA Guidance for Industry Available online: https://www.fda.gov/drugs/drug-interactions-labeling/drug-interactions-relevant-regulatory-guidance-and-policy-documents (accessed on 26 May 2023).
[6] European Medicines Agency (EMA). Guideline on the investigation of drug interactions. Available online: https://www.ema.europa.eu/en/investigation-drug-interactions-scientific-guideline (accessed on 30 May 2023).
[7] Jun, Z.; Mengmeng, J.; Lihua, Z.; Na, L.; Yonggang, L.; Zhi, S.; Xiaojian, Z.; Zhenfeng, Z. Nonlinear relationship between enteric-coated mycophenolate sodium dose and mycophenolic acid exposure in Han kidney transplantation recipients. Acta Pharmaceutica Sinica B 2017, 7, 347-352, doi:https://doi.org/10.1016/j.apsb.2016.11.003.
[8] Sobiak, J.; Resztak, M.; Banasiak, J.; Zachwieja, J.; Ostalska-Nowicka, D. High-performance liquid chromatography with fluorescence detection for mycophenolic acid determination in saliva samples. Pharmacol Rep 2023, 75, 726-736, doi:10.1007/s43440-023-00474-4.

Reviewer 2 Report
In this paper, the authors mainly studied drug-drug interactions between mycophenolic acid and trimethoprim-sulfa-methoxazole in humans, and the relationship between MPA pharmacokinetics and gut microbiota alteration. This study has certain significance for the study of medication safety after organ transplantation. The overall opinion is major revision.
Specific recommendations are as follows:
1. In this study, blood samples were drawn at pre-dose and 0.33, 0.67, 1, 1.5, 2, 4, 6, 8, 10, 12, 24, 36, and 48 hours post-MMF administration. It seems that the sampling time points are too dense for a clinical study, and it is recommended to explain.
2. The authors investigated the gut microbiota composition before and after TMP-SMX treatment, and co-occurrence network of the highly abundant genera in gut microbiota. Whether the alteration of gut microbiota was also driven by the different food intake during the study, please explain.
3. Co-occurrence network of the highly abundant genera in gut microbiota seems to be driven by the bactericidal effect of the TMP-SMX, what’s the rationale of this part? Please explain.
4. Since the correlations between gut microbiota and MPA or MPAG pharmacokinetics have been revealed, the metabolic effect of the gut microbiota on MPAG can be studied.
5. It is suggested to supplement the possible DDIs between MPA and TMP-SMX in the introduction section.
Author Response
We are thankful for the careful and insightful review and editing of our manuscript. We sincerely appreciate all the valuable comments and suggestions, which have helped us improve the quality of our manuscript. Our responses to the comments are described below in a point-by-point manner. We have incorporated the suggested changes made by the reviewers into the manuscript using Track Changes. In any case, we remain open to further comments and suggestions. Thank you for your kind consideration.
Reviewer 2’s comment #1:
In this paper, the authors mainly studied drug-drug interactions between mycophenolic acid and trimethoprim-sulfa-methoxazole in humans, and the relationship between MPA pharmacokinetics and gut microbiota alteration. This study has certain significance for the study of medication safety after organ transplantation. The overall opinion is major revision.
Specific recommendations are as follows:
- In this study, blood samples were drawn at pre-dose and 0.33, 0.67, 1, 1.5, 2, 4, 6, 8, 10, 12, 24, 36, and 48 hours post-MMF administration. It seems that the sampling time points are too dense for a clinical study, and it is recommended to explain.
Response:
We appreciate your careful consideration and feedback. For the first point, let us provide further clarification regarding the blood sampling time points in this study. The primary objective of our study was to examine the effects of TMP-SMX on MPA pharmacokinetics in humans and explore probable mechanisms underlying this pharmacokinetic DDI by focusing on the relationship between MPA pharmacokinetics and gut microbiota alteration.
Typically, the concentration-time profiles of MPA following oral administration of MMF exhibit two peaks. The maximum concentration (Cmax) of MPA varies considerably and is generally reached within 1–2 hours or less. After reaching Cmax, the MPA concentration declines rapidly over the first 4 hours and then gradually from 4 hours to 48 hours post-dose. A secondary peak often appears around 6–12 hours, which is attributed to the enterohepatic circulation of MPAG back to MPA through the action of glucuronidases from the gastrointestinal flora. The mean elimination half-life of MPA ranges from 9 to 17 hours [1-3]. These concentration-time profiles have been observed in both healthy volunteers and renal transplant recipients [3].
Considering the above information, the blood sampling time points in our study were carefully selected to cover the most important aspects of the MPA concentration-time profile. This included samples during the ascending part of the curve, around the Cmax, and more densely during the early descending phase, where dynamic changes were expected to be more pronounced.
Reviewer 2’s comment #2:
- The authors investigated the gut microbiota composition before and after TMP-SMX treatment, and co-occurrence network of the highly abundant genera in gut microbiota. Whether the alteration of gut microbiota was also driven by the different food intake during the study, please explain.
Response:
Thank you very much for pointing this out. There is increasing evidence to support the notion that diet and food intake have a significant influence on shaping the composition and function of the human gut microbiota [4]. When examining populations across different regions, dietary patterns have been found to correspond with variations in microbial composition [5]. This suggests that long-term dietary habits and patterns contribute to establishing an individual’s stable microbiota profile [6]. However, it is important to note that even with short-term and substantial changes in diet, an individual’s microbiota tends to maintain its unique and personalized composition. This suggests that factors other than diet also contribute to maintaining ecological homeostasis within the microbiota [5]. Considering that our study spanned only a 2-week duration, it is unlikely that diets had significant effects on the alteration of gut microbiota. Moreover, it is worth mentioning that on the day of the MMF intervention and pharmacokinetics study, all participants consumed the same meal.
Reviewer 2’s comment #3:
- Co-occurrence network of the highly abundant genera in gut microbiota seems to be driven by the bactericidal effect of the TMP-SMX, what’s the rationale of this part? Please explain.
Response:
Thank you for your comment. The gut microbiome is a highly complex bacterial community, and its structure is influenced by various factors, including the interactions between its members. Bacteria within the microbiome can interact with each other in diverse ways, either through targeted interactions or passively. These interactions can have beneficial, neutral, or detrimental effects on the parties involved [7]. One important aspect of microbial interactions is the concept of co-occurrence networks, which refers to the observed changes in microbial interactions and associations. These networks play a great role in shaping the structure and function of microbial communities [8]. Exploring the co-occurrence feature of microbiomes from a network perspective is very important for understanding and unraveling the role that the gut microbiota plays following drug intervention [9].
The modulating effect of TMP-SMX on the gut microbiota has been previously reported [10,11]. However, there is a lack of evidence regarding the co-occurrence patterns among microbial taxa in response to TMP-SMX. In this study, we investigated the patterns of co-occurrence genera networks and found marked differences between the pretreatment and post-TMP-SMX treatment samples (Figure 4). When comparing the post-TMP-SMX treatment samples to the pretreatment samples, we observed a noticeable increase in negative relationships among the genera. These co-occurrence analysis results were consistent with the observed reduction in β-diversity in the samples following TMP-SMX treatment. These findings imply that the co-occurrence patterns of highly abundant genera in the gut microbiota were influenced by the bactericidal effect of TMP-SMX. The antibiotic treatment seemed to disrupt the interplay between microbial taxa, leading to alterations in the co-occurrence patterns within the gut microbiota community.
Reviewer 2’s comment #4:
- Since the correlations between gut microbiota and MPA or MPAG pharmacokinetics have been revealed, the metabolic effect of the gut microbiota on MPAG can be studied.
Response:
Thank you for bringing this to our attention. Our study revealed correlations between the gut microbiota and not only MPA but also MPAG pharmacokinetics. It is known that the enterohepatic circulation of MMF involves the conversion of the stable phenolic glucuronide (MPAG) into the active form, MPA, through the action of β-glucuronidase enzymes produced by gut microbiota [1,12]. While our focus was primarily on the bacterial genera that were associated with MPA pharmacokinetics or both MPA and MPAG, we acknowledge that several other genera showed significant correlations specifically with MPAG pharmacokinetics. The abundance ratios of genera such as Muribaculaceae, Clostridia UCG-014, and an uncultured genus in the family Lachnospiraceae markedly appeared to have significantly negative correlations with the MPAG AUCs ratios, but significant positive correlations with the Vz/F ratio. This suggests that these genera may possess specific metabolic activity related to MPAG. Notably, the genus Clostridia UCG-014 and the family Lachnospiraceae have been previously associated with enriched in β-glucuronidase activity and expression, respectively [13,14]. Therefore, these genera could play a role in the metabolism of MPAG. However, further investigation is necessary to fully understand the specific mechanisms by which the gut microbiota influences MPAG pharmacokinetics.
Reviewer 2’s comment #5:
- It is suggested to supplement the possible DDIs between MPA and TMP-SMX in the introduction section.
Response:
Thank you for your comments and suggestions. We appreciate the reviewer’s input on this matter. In the revised manuscript, we have incorporated information regarding the potential DDIs between MPA and TMP-SMX in the introduction section. By including this information, we aim to provide a more comprehensive understanding of the context and significance of our study.
References
[1] Staatz, C.E.; Tett, S.E. Clinical Pharmacokinetics and Pharmacodynamics of Mycophenolate in Solid Organ Transplant Recipients. Clinical Pharmacokinetics 2007, 46, 13-58, doi:10.2165/00003088-200746010-00002.
[2] Bullingham, R.E.; Nicholls, A.J.; Kamm, B.R. Clinical pharmacokinetics of mycophenolate mofetil. Clin Pharmacokinet 1998, 34, 429-455, doi:10.2165/00003088-199834060-00002.
[3] Prémaud, A.; Debord, J.; Rousseau, A.; Le Meur, Y.; Toupance, O.; Lebranchu, Y.; Hoizey, G.; Le Guellec, C.; Marquet, P. A Double Absorption-Phase Model Adequately Describes Mycophenolic Acid Plasma Profiles in De Novo Renal Transplant Recipients Given Oral Mycophenolate Mofetil. Clinical Pharmacokinetics 2005, 44, 837-847, doi:10.2165/00003088-200544080-00005.
[4] Wastyk, H.C.; Fragiadakis, G.K.; Perelman, D.; Dahan, D.; Merrill, B.D.; Yu, F.B.; Topf, M.; Gonzalez, C.G.; Van Treuren, W.; Han, S.; et al. Gut-microbiota-targeted diets modulate human immune status. Cell 2021, 184, 4137-4153.e4114, doi:10.1016/j.cell.2021.06.019.
[5] Wu, G.D.; Chen, J.; Hoffmann, C.; Bittinger, K.; Chen, Y.Y.; Keilbaugh, S.A.; Bewtra, M.; Knights, D.; Walters, W.A.; Knight, R.; et al. Linking long-term dietary patterns with gut microbial enterotypes. Science 2011, 334, 105-108, doi:10.1126/science.1208344.
[6] Leeming, E.R.; Johnson, A.J.; Spector, T.D.; Le Roy, C.I. Effect of Diet on the Gut Microbiota: Rethinking Intervention Duration. Nutrients 2019, 11, doi:10.3390/nu11122862.
[7] Jackson, M.A.; Bonder, M.J.; Kuncheva, Z.; Zierer, J.; Fu, J.; Kurilshikov, A.; Wijmenga, C.; Zhernakova, A.; Bell, J.T.; Spector, T.D.; et al. Detection of stable community structures within gut microbiota co-occurrence networks from different human populations. PeerJ 2018, 6, e4303, doi:10.7717/peerj.4303.
[8] Kuntal, B.K.; Chandrakar, P.; Sadhu, S.; Mande, S.S. ‘NetShift’: a methodology for understanding ‘driver microbes’ from healthy and disease microbiome datasets. The ISME Journal 2019, 13, 442-454, doi:10.1038/s41396-018-0291-x.
[9] Gao, Y.; Liu, F.; Li, R.W.; Li, C.; Xue, C.; Tang, Q. Microbial Composition and Co-occurrence Patterns in the Gut Microbial Community of Normal and Obese Mice in Response to Astaxanthin. Front Microbiol 2021, 12, 671271, doi:10.3389/fmicb.2021.671271.
[10] Willmann, M.; Vehreschild, M.J.G.T.; Biehl, L.M.; Vogel, W.; Dörfel, D.; Hamprecht, A.; Seifert, H.; Autenrieth, I.B.; Peter, S. Distinct impact of antibiotics on the gut microbiome and resistome: a longitudinal multicenter cohort study. BMC Biology 2019, 17, 76, doi:10.1186/s12915-019-0692-y.
[11] Mavromanolakis, E.; Maraki, S.; Samonis, G.; Tselentis, Y.; Cranidis, A. Effect of Norfloxacin, Trimethoprim-Sulfamethoxazole and Nitrofurantoin on Fecal Flora of Women with Recurrent Urinary Tract Infections. Journal of Chemotherapy 1997, 9, 203-207, doi:10.1179/joc.1997.9.3.203.
[12] Lamba, V.; Sangkuhl, K.; Sanghavi, K.; Fish, A.; Altman, R.B.; Klein, T.E. PharmGKB summary: mycophenolic acid pathway. Pharmacogenet Genomics 2014, 24, 73-79, doi:10.1097/FPC.0000000000000010.
[13] Lopetuso, L.R.; Scaldaferri, F.; Petito, V.; Gasbarrini, A. Commensal Clostridia: leading players in the maintenance of gut homeostasis. Gut Pathogens 2013, 5, 23, doi:10.1186/1757-4749-5-23.
[14] Gloux, K.; Berteau, O.; El oumami, H.; Béguet, F.; Leclerc, M.; Doré, J. A metagenomic β-glucuronidase uncovers a core adaptive function of the human intestinal microbiome. Proceedings of the National Academy of Sciences 2011, 108, 4539-4546, doi:doi:10.1073/pnas.1000066107.
Round 2
Reviewer 1 Report
I have no additional comments to the authors.
Reviewer 2 Report
Accept in present form